# Performance of chickpea (*Cicer arietinum* L.) in maize-chickpea sequence under various integrated nutrient modules in a Vertisol of Central India

**Chetan Kumar Dotaniya**[1], **Brij Lal Lakaria**[2]*, **Yogesh Sharma**[1], **Bharat Prakash Meena**[2], **Satish Bhagwatrao Aher**[3], **Abhay Omprakash Shirale**[2], **Priya Gurav Pandurang**[2], **Mohan Lal Dotaniya**[4]*, **Ashis Kumar Biswas**[2], **Ashok Kumar Patra**[2], **Shish Ram Yadav**[1], **Madan Lal Reager**[5], **Ramesh Chandra Sanwal**[1], **Rajesh Kumar Doutaniya**[5]*, **Manju Lata**[6]

**1** Department of Soil Science & Agricultural Chemistry, Swami Keshwanand Rajasthan Agricultural University, Bikaner, India, **2** Divison of Soil Chemistry & Fertility, ICAR- Indian Institute of Soil Science, Nabibagh, Bhopal, India, **3** ICMR-National Institute for Research in Environmental Health, Bhauri, Bhopal, India, **4** Crop Production Unit, ICAR- Directorate of Rapeseed-Mustard Research, Sewar, Bharatpur, India, **5** Department of Agronomy, Sri Karan Narendra Agriculture University, Jobner, Jaipur, Rajasthan, India, **6** Rajasthan University, Jaipur, India

* lakaria2001@gmail.com (BLL); rajeshkumardoutaniya@gmail.com (RKD); mohan30682@gmail.com (MLD)

**Data Availability Statement:** All relevant data are within the paper.

## Abstract

Present investigation was conducted at the Research Farm of Indian Institute of Soil Science, Bhopal during 2017–18 and 2018–19 to study the performance of chickpea crop under various nutrient management modules in a Vertisol. The field experiment was set up in a randomized block design with three replications of twelve different INM modules. During the *rabi* seasons of 2017–18 and 2018–19, the chickpea (cv. JG-315) was grown with a set of treatments. The crop's performance was evaluated in terms of growth, yield (grain and straw), nutritional content, and nutrient uptake under different treatments. At crop harvest, the physic-chemical characteristics of the soil were also evaluated. Finally, the relationship between the numerous examined parameters was determined. The results showed that integrated nutrient management modules had a positive impact on chickpea crop performance and productivity when compared to using only inorganic fertilizer. The INM modules dramatically increased soil organic carbon and improved soil health in terms of physical and chemical qualities, in addition to higher crop performance. Among the various modules, (1) application of 75% STCR dose + FYM @ 5t ha$^{-1}$ to maize followed by 100% P only to chickpea and (2) application of FYM @ 20t ha$^{-1}$ to maize followed by FYM @ 5t ha$^{-1}$ to chickpea increased the productivity and nutrient uptake in chickpea, improved soil physico-chemical properties and reflected as viable technique in improving soil nutrient availability on sustainable basis.

**Funding:** The study was financially supported by University Grants Commission during the Ph.D degree (grant no. RGNF-2017-18-Raj-44310), awarded to CKD.

## Introduction

As a result of rapid urbanization and industrialization, India is facing a dire predicament in which the number of mouths to feed will continue to rise at an alarming rate while available agricultural land area remains inelastic, if not declining. As a result, adopting intense cropping is unavoidable in order to increase food grain output. Second, because of the quick visual impacts of nitrogenous fertilizer on plants and the higher expense of phosphatic and potassic fertilizers, most farmers prefer to apply solely nitrogenous fertilizer to crops, resulting in an imbalance of nutrient consumption and nutrient depletion from the soil. High crop yields can be maintained by judicious application of manures and fertilizers. Crop rotation with cereals and legumes, as well as organic manure addition and optimal N P K application to the soil system, are crucial for crop yield and soil health [1, 2]. Maize-chickpea rotation is an important cropping sequence of India [3]. Maize is an important cereal crop in the global agricultural economy, serving as both human food and animal feed. Maize, after sugar cane, is the most demanding crop in this cropping sequence, requiring both micro and macro nutrients throughout its growing period to achieve high growth and yield potentials.

Long-term sustainability concerns are rising in agriculture as a result of over- and under-application of fertilizers, as well as inadequate resource management, which is causing soil health to deteriorate and crop production to decline [4–6]. The most obvious notion for regulating and maintaining soil health and crop productivity is the integrated use of organic and inorganic sources of nutrients [7, 8]. Integrated nutrient management is one of the most important components towards enhancing crop production and sustaining soil fertility [4, 5, 9]. Due to application of P fertilizers during crop production, a huge amount of P converts into unavailable forms and is fixed into the soil [10]. The use of locally available organic residues (OR), *i.e.* bagasse, press mud and rice straw can be a cheaper technology to mobilize unavailable P in situ [11]. The addition of organic material increased the amount of food available to microorganisms, allowing for faster breakdown. The presence of sugar in organic wastes promotes decomposition and increases the release of low molecular organic acids into the soil [8, 12]. Application of sugarcane industries by-products reduces the RDF and improves organic matter of soil during the crop production [13]. Chickpea is a popular pulse crop farmed and consumed throughout the world, particularly in Afro-Asian countries. It is also one of the major pulse crops cultivated and consumed in India. Chickpea recorded a highest ever production of 11.23 Mt at a record productivity level of 1063 kg ha$^{-1}$ in an area of 10.56 M ha. In India, among 7 major chickpea producing states MP ranks 4.60 Mt and contribute more than 40 per cent of total chickpea production in India. Keeping in view the above the present study was undertaken to study the performance of chickpea under various INM modules.

## Materials and methods

### Experiment site

A field experiment was conducted at the research farm of the ICAR-Indian Institute of Soil Science, Bhopal. Geographically, the ICAR-IISS lies between 23˚18' N latitude and 77˚24' E longitudes. The elevation above mean sea level is 485 m. The climate of study site is sub-humid and in characterized by cool and dry winters, hot summer and humid monsoon with a mean annual rainfall of 1146 mm 75–80% of which is received during June to September.

### Experiment details

The Chickpea (JG-315) was grown with 70 kg ha$^{-1}$ seed rate with row to row and plant to plant spacing of 30cm X 10 cm in a Vertisol at the research farm of the Indian Institute of Soil Science with 15 different treatments (Table 1). The general recommended dose (GRD) of

**Table 1. Treatment details in maize-chickpea cropping sequence.**

| Treatment | Maize | Chickpea |
|---|---|---|
| T$_1$ (Control) | No Fertilizer/ Manure | No Fertilizer/ Manure |
| T$_2$ (GRD) | 120–60–30 | 20-60-20 |
| T$_3$ (STCR) | 135-55-50 (5 t ha$^{-1}$) | 0-0-0(1.5 t ha$^{-1}$) |
| T$_4$ | 75% NPK of T$_3$ | 100% P only |
| T$_5$ | 75% NPK of T$_3$ + 5 t ha$^{-1}$ FYM | 100% P only |
| T$_6$ | 75% NPK of T$_3$ + 1 t ha$^{-1}$ PM | 100% P only |
| T$_7$ | 75% NPK of T$_3$ + 5 t ha$^{-1}$ UC | 100% P only |
| T$_8$ | 75% NPK of T$_3$ + MR incorporated | 100% P only + MR mulch |
| T$_9$ | 1 t ha$^{-1}$ PM + Gly 2 t ha$^{-1}$ + MR incorporated | 100% P only + MR mulch |
| T$_{10}$ | 5 t ha$^{-1}$ FYM + Gly 2 t ha$^{-1}$ + MR incorporated | 100% P only + MR mulch |
| T$_{11}$ | 20 t ha$^{-1}$ FYM (every season) | 5 t ha$^{-1}$ FYM (Every Season) |
| T$_{12}$ | 75% NPK of T$_3$ + 20 t ha$^{-1}$ FYM (once in 4 years) | 100% P only |

GRD—General recommended dose (kg ha$^{-1)}$, STCR—Soil test crop response dose, MR—Maize residues, FYM—Farm yard manure, PM—Poultry manure, UC—Urban compost, WR- Wheat Residue, Gly–Glyricidia.

fertilizer application for chickpea was 20-60-20 kg N, P$_2$O$_5$ and K$_2$O, respectively. The plot size of the experiment was 20 X 5 sq meters.

## Experimental soil

The soil of experimental site is classified as Vertisol (*Typic Haplusterts*) with smectite as the dominant clay mineral. Vertisols are churning heavy clay soils with a high proportion of swelling clays. These soils form deep wide cracks during summer season. The soil of the experimental site is clayey in texture with 25.2, 18 and 56.8 per cent of sand, silt and clay, respectively. The soil was medium in soil organic carbon (0.53%), low in available N (68.8 mg kg$^{-1}$), medium in available P (12.8 mg kg$^{-1}$) and high in available K (237 mg kg$^{-1}$). The soil was normal in reaction (pH 7.76) and electrical conductivity (EC) was 0.48 dS m$^{-1}$.

## Crop performance, nutrient concentration and uptake

The performance of the chickpea crop was evaluated by monitoring the growth and yield parameters viz., plant height, seed index (100 seed), seed yield and straw yield. The plant samples were collected at harvest stage and thoroughly washed with distilled water, dried at room temperature for 24 hours; oven dried at 65˚C till constant weight. The samples were ground to a homogenous powder using grinding machine. The samples were digested in H$_2$SO$_4$ and di-acid (HNO$_3$: HClO$_4$ at 9:4 ratio) for estimation of N and; P and K, respectively [14]. The N, P and K in digest were determined by modified Kjeldhal method, vandomolybdate phosphoric yellow colour method and flame photometer method, respectively [15]. The uptake of the nutrients was calculated from the concentration and respective seed/straw yield. The total uptake was calculated by combining straw and seed uptake.

## Soil sampling, processing and analysis

Soil samples were collected from 0–15 cm depth after harvest of *rabi* season each year and were homogenized. Visible litter and roots were picked out from collected soil samples. The

soil samples were air-dried at room temperature, ground and sieved through 2 mm sieve. The processed soil was stored in air tight plastic containers for further analysis of different properties. The soil pH was measured in soil water suspension (1:2.5 ratio) using pH meter [15]. The soil water suspension used for pH estimation was kept for one hour, the electrical conductivity (dS m$^{-1}$) was measured in the supernatant using electrical conductivity meter after standardizing the instrument with 0.01 $M$ KCl. The soil organic carbon content (%) in soil samples was determined in soil using the wet oxidation methods [16]. Available nitrogen content in soil was estimated by alkaline permanganate method [17] in which a known quantity of soil was reacted with 0.32% alkaline KMNO$_4$ and 2.5% NaOH in a Kjeldhal flask and the contents were distilled. The liberated ammonia was absorbed in boric acid containing mixed indicator and N was estimated by titrating the distillate against standard sulphuric acid till bluish green colour changed to original color boric acid. The available phosphorus was determined using Olsen's method (0.5 $N$ NaHCO$_3$ with pH 8.5). The phosphorus content extract was determined by ascorbic acid method [18]. The soil available potassium was extracted from the soil using neutral normal ammonium acetate in soil solution (ratio 1:5) and the K in extract was recorded using flame photometer [19].

## Statistical analysis

The experiment comprised with twelve treatments laid out in a randomized block design (RBD) with three replications. All the measurements are the mean value of three separate replicates. Data was subjected to an analysis of variance. The mean values were grouped for comparisons and the least significant differences among them were calculated at $P < 0.05$ confidence level using ANNOVA statistics as outlined by [20].

## Results

### Growth and yield of chickpea

During the experimental period, the performance of the chickpea crop was measured in terms of growth and yield parameters, grain yield, and biomass. The plant height of chickpea was recorded at 30 DAS, 60 DAS and 90 DAS, respectively during the study years *viz*., 2017–18 and 2018–19 (Table 2). The results revealed that the plant height of chickpea recorded at periodic interval during the study period found significantly higher under the treatment receiving FYM@ 20 t ha$^{-1}$ (T$_{11}$) and the treatment receiving STCR recommended dose of fertilizers (T$_3$) as compared to the other treatments. The seed index of chickpea found significantly influenced under the integrated nutrient application (Table 3). The pooled data of two years indicated highest seed index under the treatment receiving FYM@ 20 t ha$^{-1}$ followed by the treatment with STCR recommended dose of fertilizers. The grain yield of chickpea ranged 1.37–2.39 t ha$^{-1}$, 1.05–2.15 t ha$^{-1}$ and 1.21–2.27 t ha$^{-1}$ during 2017–18, 2018–19 and pooled of two years, respectively. The average grain yield of chickpea across the treatments was found 1.97 t ha$^{-1}$, 1.66 t ha$^{-1}$ and 1.81 t ha$^{-1}$ during 2017–18, 2018–19 and pooled of two years, respectively. The pooled data indicated that the treatment receiving FYM @ 5t ha$^{-1}$ (T$_{11}$) recorded highest grain yield (2.27 t ha$^{-1}$) and found statistically significant over rest of the treatments.

### Nutrient accumulation in chickpea seed and straw

The chickpea grain N concentration ranged 2.67–2.96%, 2.64–2.90% and 2.65–2.93% during 2017–18, 2018–19 and pooled of two years, respectively. Similarly, the average chickpea grain N concentration across the treatments was found 2.79% during the study period. The chickpea grain P concentration ranged 0.40–0.49%, 0.41–0.50% and 0.41–0.50% during 2017–18, 2018–

**Table 2. Plant height of chickpea at different growth stages under various INM modules.**

| Treatments | Plant Height (cm) | | | | | | | | |
|---|---|---|---|---|---|---|---|---|---|
| | 30 DAS | | | 60 DAS | | | 90 DAS | | |
| | 2017–18 | 2018–19 | Pooled | 2017–18 | 2018–19 | Pooled | 2017–18 | 2018–19 | Pooled |
| $T_1$ | 11.2 | 13.1 | 12.2 | 31.8 | 33.6 | 32.7 | 45.7 | 48.1 | 46.9 |
| $T_2$ | 12.4 | 14.3 | 13.3 | 34.9 | 36.8 | 35.9 | 52.6 | 54.7 | 53.6 |
| $T_3$ | 13.4 | 15.7 | 14.5 | 36.3 | 38.1 | 37.2 | 57.0 | 63.7 | 60.4 |
| $T_4$ | 12.6 | 14.2 | 13.4 | 33.9 | 35.7 | 34.8 | 51.3 | 54.7 | 53.0 |
| $T_5$ | 12.8 | 14.6 | 13.7 | 34.7 | 36.6 | 35.7 | 51.4 | 57.3 | 54.4 |
| $T_6$ | 11.8 | 13.6 | 12.7 | 33.6 | 35.4 | 34.5 | 50.9 | 54.3 | 52.6 |
| $T_7$ | 12.2 | 14.1 | 13.1 | 33.8 | 35.6 | 34.7 | 49.1 | 55.1 | 52.1 |
| $T_8$ | 11.6 | 13.4 | 12.5 | 33.7 | 35.5 | 34.6 | 48.9 | 54.8 | 51.8 |
| $T_9$ | 12.1 | 14.0 | 13.1 | 33.0 | 34.9 | 34.0 | 49.7 | 53.3 | 51.5 |
| $T_{10}$ | 12.2 | 14.1 | 13.2 | 35.6 | 37.3 | 36.5 | 54.2 | 59.2 | 56.7 |
| $T_{11}$ | 13.6 | 17.5 | 15.6 | 40.6 | 42.5 | 41.5 | 59.6 | 69.2 | 64.4 |
| $T_{12}$ | 12.9 | 15.1 | 14.0 | 34.8 | 37.3 | 36.1 | 51.0 | 59.8 | 55.4 |
| SEm± | 0.49 | 0.31 | 0.38 | 1.10 | 1.02 | 1.06 | 2.94 | 1.81 | 1.79 |
| CD (p = 0.05) | 1.44 | 0.91 | 1.10 | 3.23 | 2.98 | 3.09 | 8.64 | 5.31 | 5.23 |

19 and pooled of two years, respectively (Table 4). Similarly, the average chickpea grain P concentration across the treatments was found 0.44% and 0.45% during 2017–18 and 2018–19, respectively. The chickpea grain K concentration ranged 1.01–1.14%, 1.05–1.15% and 1.03–1.14% during 2017–18, 2018–19 and pooled of two years, respectively. Similarly, the average chickpea grain K concentration across the treatments was found 1.10%, 1.12% and 1.11% during 2017–18, 2018–19 and pooled of two years, respectively. The chickpea stover N concentration ranged 0.52–0.68%, 0.55–0.73% and 0.54–0.71% during 2017–18, 2018–19 and pooled of two years, respectively. Similarly, the average chickpea stover N concentration across the treatments was found 0.63%, 0.67% and 0.65% during 2017–18, 2018–19 and pooled of two years. The chickpea stover P concentration ranged 0.11–0.14%, 0.10–0.15% and 0.10–0.14% during 2017–18, 2018–19 and pooled of two years, respectively. Similarly, the average chickpea stover

**Table 3. Chickpea seed index, seed and straw yield.**

| Treatments | Seed Index (g 100 seed$^{-1}$) | | | Seed Yield (t ha$^{-1}$) | | | Straw Yield (t ha$^{-1}$) | | |
|---|---|---|---|---|---|---|---|---|---|
| | 2017–18 | 2018–19 | Pooled | 2017–18 | 2018–19 | Pooled | 2017–18 | 2018–19 | Pooled |
| $T_1$ | 12.47 | 12.70 | 12.58 | 1.37 | 1.05 | 1.21 | 2.13 | 1.59 | 1.86 |
| $T_2$ | 13.70 | 13.80 | 13.75 | 2.02 | 1.58 | 1.80 | 3.08 | 2.41 | 2.74 |
| $T_3$ | 13.93 | 14.13 | 14.03 | 2.03 | 1.66 | 1.85 | 2.79 | 2.44 | 2.61 |
| $T_4$ | 13.23 | 13.23 | 13.23 | 1.77 | 1.42 | 1.60 | 2.55 | 2.26 | 2.40 |
| $T_5$ | 13.43 | 13.30 | 13.37 | 1.89 | 1.70 | 1.80 | 2.85 | 2.68 | 2.76 |
| $T_6$ | 13.33 | 13.07 | 13.20 | 1.95 | 1.87 | 1.91 | 2.99 | 3.17 | 3.08 |
| $T_7$ | 13.40 | 13.57 | 13.48 | 2.22 | 1.59 | 1.91 | 3.35 | 3.02 | 3.18 |
| $T_8$ | 13.27 | 13.33 | 13.30 | 1.74 | 1.23 | 1.48 | 2.77 | 2.43 | 2.60 |
| $T_9$ | 13.17 | 13.27 | 13.22 | 2.10 | 2.00 | 2.05 | 3.01 | 3.26 | 3.13 |
| $T_{10}$ | 13.47 | 13.47 | 13.47 | 1.83 | 1.66 | 1.74 | 2.61 | 3.03 | 2.82 |
| $T_{11}$ | 14.33 | 14.93 | 14.63 | 2.39 | 2.15 | 2.27 | 3.49 | 3.78 | 3.63 |
| $T_{12}$ | 13.57 | 13.53 | 13.55 | 2.28 | 1.97 | 2.12 | 3.33 | 3.34 | 3.33 |
| SEm± | 0.25 | 0.27 | 0.20 | 0.11 | 0.08 | 0.08 | 0.17 | 0.23 | 0.17 |
| CD (p = 0.05) | 0.72 | 0.78 | 0.58 | 0.31 | 0.23 | 0.22 | 0.48 | 0.66 | 0.50 |

**Table 4. Nutrient concentration in chickpea under various INM modules.**

| Treatments | Seed | | | Straw | | |
|---|---|---|---|---|---|---|
| | N (%) | P (%) | K (%) | N (%) | P (%) | K (%) |
| $T_1$ | 2.04 | 0.30 | 1.03 | 0.54 | 0.10 | 1.40 |
| $T_2$ | 2.20 | 0.34 | 1.13 | 0.65 | 0.13 | 1.47 |
| $T_3$ | 2.26 | 0.37 | 1.14 | 0.68 | 0.14 | 1.48 |
| $T_4$ | 2.08 | 0.32 | 1.06 | 0.64 | 0.12 | 1.42 |
| $T_5$ | 2.22 | 0.36 | 1.13 | 0.70 | 0.14 | 1.48 |
| $T_6$ | 2.10 | 0.33 | 1.11 | 0.64 | 0.13 | 1.46 |
| $T_7$ | 2.16 | 0.34 | 1.08 | 0.67 | 0.12 | 1.44 |
| $T_8$ | 2.10 | 0.31 | 1.07 | 0.63 | 0.12 | 1.43 |
| $T_9$ | 2.14 | 0.32 | 1.12 | 0.65 | 0.12 | 1.43 |
| $T_{10}$ | 2.18 | 0.34 | 1.13 | 0.67 | 0.13 | 1.48 |
| $T_{11}$ | 2.21 | 0.36 | 1.14 | 0.71 | 0.13 | 1.48 |
| $T_{12}$ | 2.12 | 0.34 | 1.12 | 0.66 | 0.12 | 1.45 |
| SEm± | 0.05 | 0.01 | 0.02 | 0.02 | 0.01 | 0.02 |
| CD (p = 0.05) | 0.14 | 0.04 | 0.06 | 0.05 | 0.02 | 0.04 |

P concentration across the treatments was found 0.13% during the study period. The chickpea stover K concentration ranged 1.40–1.49% during the study period 2017–2019. Similarly, the average chickpea stover K concentration across the treatments was found 1.45–1.46%.

## Nutrient uptake

The chickpea total N uptake ranged 47.72–92.24 kg ha[-1], 36.45–89.83 kg ha[-1] and 42.09–91.04 kg ha[-1] during 2017–18, 2018–19 and pooled of two years, respectively (Fig 1). Similarly, the average chickpea total N uptake across the treatments was found 73.77 kg ha[-1], 65.21 kg ha[-1] and 69.49 kg ha[-1] during 2017–18, 2018–19 and pooled of two years, respectively. The chickpea total P uptake ranged 9.99–20.12 kg ha[-1] during the study period.

Similarly, the average chickpea total P uptake across the treatments was found between 7.53–20.93 and 8.76–20.52 kg ha[-1] during 2017–18, 2018–19 and pooled of two years (Fig 2). The chickpea total K uptake ranged 52.7–157.7 kg ha[-1], 53.5–137.0 kg ha[-1] and 53.1–147.3 kg ha[-1] during 2017–18, 2018–19 and pooled of two years, respectively.

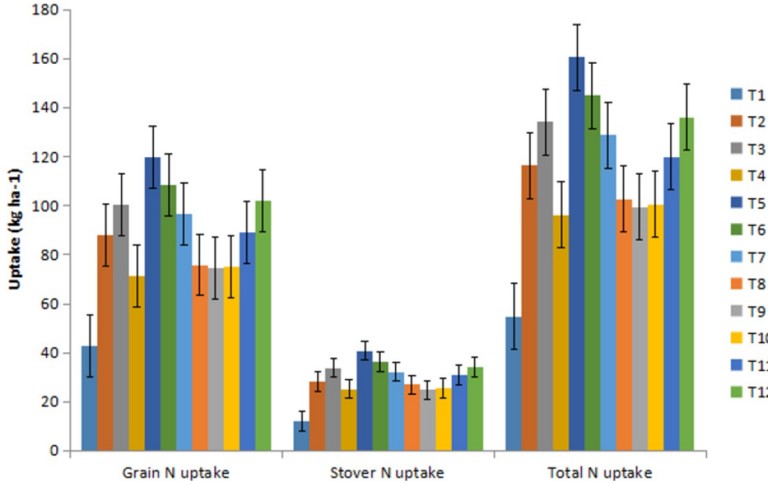

**Fig 1. Nitrogen uptake in chickpea under various INM modules.**

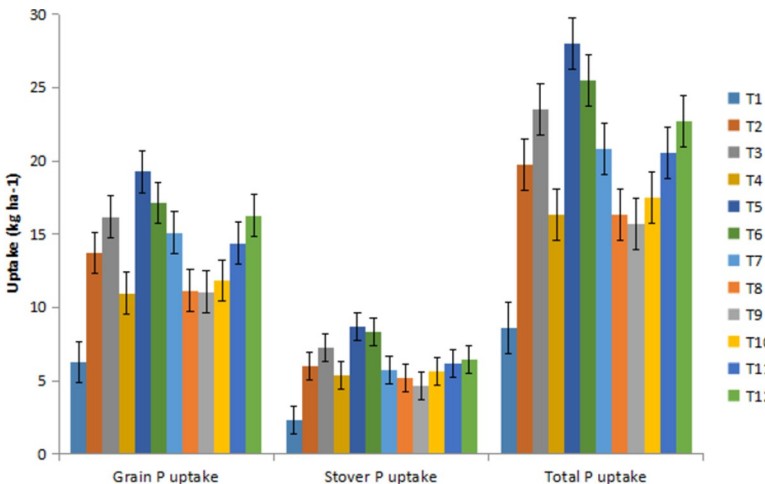

**Fig 2. Phosphorous uptake in chickpea under various INM modules.**

Similarly, the average chickpea total K uptake across the treatments was found 115.0 kg ha$^{-1}$, 104.0 kg ha$^{-1}$ and 109.5 kg ha$^{-1}$ during 2017–18, 2018–19 and pooled of two years, respectively (Fig 3). In general, the uptake of N, P and K in chickpea was found higher under the treatments receiving the integrated nutrient management modules/recommended dose of balanced chemical fertilizers (STCR).

## Soil properties under various INM modules

**Soil pH and EC.**   The soil pH of the soil ranged 7.83–7.89 and 7.85–7.89 in 0–15 cm and 15–30 cm soil depth respectively. Similarly, the average pH value in 0–15 cm and 15–30 cm was recorded as 7.86 and 7.87, respectively across the various INM treatments under study. Similarly, the EC of the soil ranged 0.22–0.26 dS m$^{-1}$ and 0.23–0.36 dS m$^{-1}$ in 0–15 cm and 15–30 cm soil depth, respectively.

**Soil organic carbon (SOC).**   The soil organic carbon ranged 0.60–0.89% and 0.42–0.50% in 0–15 cm and 15–30 cm soil depth, respectively under various treatments under study. Similarly, the average SOC across the different treatments in 0–15 cm and 15–30 cm soil depth was found 0.69% and 0.45%, respectively. The highest SOC (0.89%) was found under the treatment receiving FYM @ 20 t ha$^{-1}$ to maize followed by FYM @ 5 t ha$^{-1}$ to chickpea ($T_{11}$). The

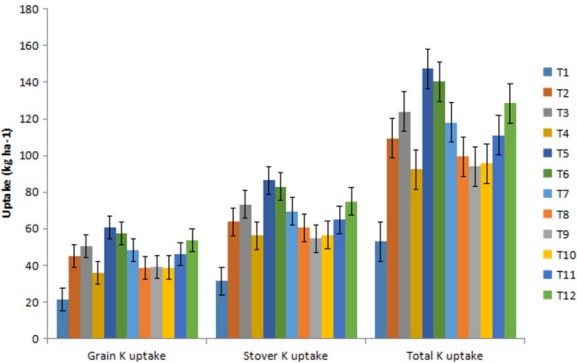

**Fig 3. Potassium uptake in chickpea under various INM modules.**

**Table 5. The effect of various INM modules on soil properties (0–15 cm) at chickpea harvest.**

| Treatment | pH | EC | OC | N | P | K |
|---|---|---|---|---|---|---|
| $T_1$ | 7.83 | 0.22 | 0.60 | 201.0 | 9.1 | 275.8 |
| $T_2$ | 7.85 | 0.24 | 0.65 | 262.7 | 25.2 | 305.8 |
| $T_3$ | 7.86 | 0.25 | 0.63 | 315.4 | 30.5 | 377.2 |
| $T_4$ | 7.85 | 0.23 | 0.61 | 256.4 | 19.5 | 340.2 |
| $T_5$ | 7.84 | 0.24 | 0.73 | 273.4 | 28.0 | 357.5 |
| $T_6$ | 7.87 | 0.25 | 0.66 | 261.1 | 25.0 | 340.1 |
| $T_7$ | 7.88 | 0.26 | 0.69 | 246.3 | 22.8 | 369.3 |
| $T_8$ | 7.86 | 0.26 | 0.68 | 283.4 | 20.7 | 351.4 |
| $T_9$ | 7.89 | 0.25 | 0.69 | 281.2 | 19.7 | 351.1 |
| $T_{10}$ | 7.85 | 0.24 | 0.73 | 286.9 | 24.6 | 341.7 |
| $T_{11}$ | 7.87 | 0.25 | 0.89 | 344.1 | 32.6 | 430.9 |
| $T_{12}$ | 7.84 | 0.23 | 0.71 | 282.2 | 29.2 | 363.7 |
| SEm± | 0.04 | 0.02 | 0.02 | 12.2 | 1.2 | 20.8 |
| CD | NS | NS | 0.07 | 35.6 | 3.4 | 61.0 |

treatments consist of organic manures as full or partial input, i.e. $T_5$, $T_7$, $T_8$, $T_9$, $T_{10}$ and $T_{12}$ showed significantly higher SOC.

**Soil available N, P and K.** The soil available N concentration in 0–15 cm soil layer ranged 195.9–350.4 kg ha$^{-1}$, 206.1–337.9 kg ha$^{-1}$ and 201.0–344.1 kg ha$^{-1}$ during 2017–18, 2018–19 and pooled of two years, respectively. Similarly, the average soil available N concentration across the treatments was found 282.8 kg ha$^{-1}$, 266.2 kg ha$^{-1}$ and 274.5 kg ha$^{-1}$ during 2017–18, 2018–19 and pooled of two years, respectively. The pooled data of two years with respect to soil available N revealed that the highest value was observed in the treatment $T_{11}$ (FYM @ 20 t ha$^{-1}$ to maize followed by FYM @ 5t ha$^{-1}$ to chickpea). The soil available P concentration in 0–15 cm soil layer ranged 9.3–31.0 kg ha$^{-1}$, 8.8–34.2 kg ha$^{-1}$ and 9.1–32.6 kg ha$^{-1}$ during 2017–18, 2018–19 and pooled of two years, respectively. Similarly, the average soil available P concentration across the treatments was found 21.8 kg ha$^{-1}$, 26.0 kg ha$^{-1}$ and 23.9 kg ha$^{-1}$ during 2017–18, 2018–19 and pooled of two years, respectively. The soil available P was found highest in the treatment $T_{11}$ (FYM @ 20 t ha$^{-1}$ to maize followed by FYM @ 5t ha$^{-1}$ to chickpea). The soil available K under various long term INM modules ranged 285.7–416.8 mg kg$^{-1}$ (Table 5). Similarly, the average soil available K across the treatments under study was found 332.8 mg kg$^{-1}$. The data revealed that the treatment receiving FYM @ 20 t ha$^{-1}$ to maize followed by FYM @ 5t ha$^{-1}$ ($T_{11}$) showed highest soil available K (416.8 mg ka$^{-1}$) and found statistically significant over rest all the treatments under study. The treatment receiving STCR dose ($T_3$), 75% STCR dose + FYM @ 5t ha$^{-1}$ ($T_5$), 75% STCR dose + maize residue mulching ($T_8$), 1 t PM + Gly 2 t + maize residue mulching ($T_9$), 5 t FYM + Gly 2 t + maize residue mulching ($T_{10}$) and 75% STCR dose + 20 t FYM once in four years ($T_{12}$) found statistically at par. Similarly, the treatments $T_2$ (GRD), $T_4$ (75% of STCR dose), $T_6$ (75% of STCR dose + 1 t PM) and $T_7$ (75% of STCR dose + 5 t VC) were also found statistically similar with respect to the soil available K. The treatment control ($T_1$) showed lowest value of soil available K (332.8 mg kg$^{-1}$).

## Discussion

Chickpea growth and yield were increased by combining chemical and organic plant nutrients. The treatments receiving integrated nutrient management incorporating the application of chemical fertilisers and manures ($T_6$, $T_7$, $T_9$, and $T_{12}$) were shown to be either superior or comparable to those receiving only chemical fertiliser. The better performance of crop with

adequate and balanced chemical fertilizer application has already been documented [21–27]. However, the imbalanced use of chemical fertilizers deteriorating the soil health. Similarly, the supply and cost of the chemical fertilizers limits its access and application. The use of integrated nutrient management hence introduced to substitute the partial chemical fertilizer requirement. The integrated nutrient management modules in present investigation proved at par and even better compared to the sole chemical fertilizer application with respect to the performance of the chickpea crop. Dotaniya et al. [9] reported increased seed and straw yield under the application of integrated nutrient. Similarly Srinivasa et al. [28] also found application of recommended N, $P_2O_5$ with 40 kg $K_2O$ ha$^{-1}$ + 6 t of FYM recorded higher grain yield (2266.07 kg ha$^{-1}$) and straw yield (3814.95 kg ha$^{-1}$) of foxtail millet in red sandy loam soils.

A long- term application of FYM and inorganic fertilizers that the grain yield and uptake of N, P and K by both crops were higher with the application of FYM and inorganic fertilizer than in control plots. The uptake of N, P and K increased with the application of FYM and $N_{100}$, $P_{50}$, $K_{50}$ [29]. Shivran et al. [30] recorded that application of FYM @ 5t ha$^{-1}$, inorganic $P_2O_5$ @ 40 kg ha$^{-1}$ and S @ 20 kg ha$^{-1}$ resulted significantly higher plant height, branchesplant$^{-1}$, number of podsplant$^{-1}$, number of seedspod$^{-1}$, number of nodules, dry weight of nodules and hence higher seed, straw yield and protein content of soybean during 2008 and 2009. There were no significant difference observed within 40 and 60 kg $P_2O_5$ ha$^{-1}$. Similarly, Balai et al. [31] also reported that N content and uptake in grain increased significantly with increasing levels of phosphorus up to 60 kg ha$^{-1}$ but N content and uptake in straw increased significantly up to 40 kg $P_2O_5$ ha$^{-1}$. P content and uptake by grain and straw was significantly increased by applying increasing level of phosphorus up to 60 and 40 kg $P_2O_5$ ha$^{-1}$ in chickpea crop. Asangi et al. [32] reported nitrogen (1.48 and 0.69%), phosphorus (0.44 and 0.26%), potassium (0.63 and 1.97%) in maize grain and stover, respectively. Malvi et al. [33] Reported highest maximum N, P, K and S nutrient uptakes in fodder berseem plants under 90 kg $K_2O$ and 40 kg S ha$^{-1}$ application. The results of present investigation are in good agreement with these findings.

In general, the concentration of N, P and K in chickpea (grain and straw) was found higher under the treatments receiving the integrated nutrient management modules/recommended dose of balanced chemical fertilizers (STCR). Shivran et al. [30] recorded that application of FYM @ 5t ha$^{-1}$, inorganic $P_2O_5$ @ 40 kg ha$^{-1}$ and S @ 20 kg ha$^{-1}$ resulted significantly higher plant height, branches plant$^{-1}$, number of pods plant$^{-1}$, number of seeds pod$^{-1}$, number of nodules, dry weight of nodules and hence higher seed, straw yield and protein content of soybean during 2008 and 2009. There were no significant difference observed within 40 and 60 kg $P_2O_5$ha$^{-1}$. Balai et al. [31] reported that N content and uptake in grain increased significantly with increasing levels of phosphorus up to 60 kg ha$^{-1}$ but N content and uptake in straw increased significantly up to 40 kg $P_2O_5$ ha$^{-1}$. P content and uptake by grain and straw was significantly increased by applying increasing level of phosphorus up to 60 and 40 kg $P_2O_5$ ha$^{-1}$ in chickpea crop. Asangi et al. [32] reported nitrogen (1.48 and 0.69%), phosphorus (0.44 and 0.26%), potassium (0.63 and 1.97%) in maize grain and stover, respectively. Malvi et al. [34] resulted highest maximum N, P, K and S contents in berseem plants under 90 kg $K_2O$ and 40 kg S ha$^{-1}$ application.

The type of organic inputs as well as the physico-chemical qualities of soils has influenced soil attributes. Similarly, the average EC value in 0–15 cm and 15–30 cm was recorded as 0.24 dS m$^{-1}$ and 0.28 dS m$^{-1}$, respectively across the various INM treatments under study. Both, pH and soil EC did not reveal any significant change under long term application of the various INM modules [35, 36]. Yadav et al. [37] had also recorded non-significant changes in soil pH and EC with the application of chemical fertilizers and organic manures which may be attributed to the fact that the soil pH is mainly affected by the parent material involved in soil formation and the climatic conditions [4, 17].

Soil organic carbon served as a source of food for soil microbial biomass and moderated plant nutrient dynamics in the soil. Manna *et al.* [38] also found increase in SOC with the application of FYM alone or in combination with recommended NPK fertilizers over absolute control and sole NPK fertilizer application [2, 39, 40]. The results of this study are in close agreement with these findings. The higher C accumulation in the Vertisol may be attributed to their high silt+clay content which increase the C stabilization capacity [41]. The findings of [42–46] with respect to the soil organic carbon are in line with the results of present investigation.

The soil available N was positively impacted by the various long-term INM modules. This was probably due to the mineralization of N from added organic manure (FYM) [47]. Upon addition of organic matter, the available nutrient status of soil increases considerably due to mineralization from soil as well as its own nutrient contents [37]. The higher N content under INM application are in conformity with the finding of who reported increase in soil available N due to FYM application [48]. Similarly, it was also found that higher soil available N under the sole/combined application of organic manures [49, 50]. The influence of various long-term INM modules significantly influenced the soil available P as the values showed treatment wise variation. The increased availability of P in soil upon organic application is mainly attributed to the release of inorganic P from added organics, inhibition of P adsorption by organic molecules released from the organics, a rise in soil pH during decomposition and complexation of soluble $Al^{3+}$ and Fe by organic molecules [20, 51]. Aziz *et al.* [49] reported maximum soil N and P contents after a maize harvest for FYM whereas the minimum N and P contents were found for the treatments with the application of inorganic NPK fertilizer. Kong *et al.* [52] found that availability of soil N, P and K was considerably decline under control and the treatment that received only chemical fertilizers. This indicates that decreasing N, P and K availability in soil could be a threat to long-term sustainability of crop productivity and soil health [53] Furthermore, Yang et al. [54] observed that integration of FYM with 75% NPK of SCTR improved the SOC content in surface soil over the initial value or unfertilized plots. A similar trend was observed for P and K availability in soil system, indicating that significantly build up the Olsen P in surface soil with continuous used of FYM (20 Mg ha$^{-1}$) and integration of 75% NPK along with FYM as compared to rest of INM practices. Many researchers have indicated that the application of organic and inorganic substrats during crop production has a significant impact on accessible macronutrients in soil [7, 55, 56].

## Conclusion

The integrated nutrient management modules positively influenced the performance and productivity of chickpea crop as compared to the sole inorganic fertilizer application. Besides the superior crop performance, the INM modules significantly enhanced soil organic carbon and improved soil health in terms of soil physical and chemical properties. Among the various modules, (1) application of 75% STCR dose + FYM @ 5t ha$^{-1}$ to maize followed by 100% P only to chickpea and (2) application of FYM @ 20t ha$^{-1}$ to maize followed by FYM @ 5t ha$^{-1}$ to chickpea increased the productivity and nutrient uptake in chickpea, improved soil physicochemical properties and reflected as viable technique in improving soil nutrient availability on sustainable basis.

## Acknowledgments

The authors are thankful to the Director, ICAR-Indian Institute of Soil Science, Bhopal and Vice Chancellor, Swami Keshwanand Rajasthan Agricultural University, Bikaner for providing the research opportunity and for extending every kind of support during the study.

## Author Contributions

**Conceptualization:** Chetan Kumar Dotaniya, Brij Lal Lakaria, Yogesh Sharma, Mohan Lal Dotaniya.

**Data curation:** Bharat Prakash Meena, Shish Ram Yadav.

**Formal analysis:** Bharat Prakash Meena, Satish Bhagwatrao Aher, Mohan Lal Dotaniya, Shish Ram Yadav, Ramesh Chandra Sanwal, Rajesh Kumar Doutaniya.

**Investigation:** Chetan Kumar Dotaniya, Brij Lal Lakaria.

**Methodology:** Abhay Omprakash Shirale, Priya Gurav Pandurang, Rajesh Kumar Doutaniya.

**Project administration:** Brij Lal Lakaria, Ashis Kumar Biswas, Ashok Kumar Patra.

**Resources:** Abhay Omprakash Shirale.

**Software:** Satish Bhagwatrao Aher, Shish Ram Yadav.

**Supervision:** Yogesh Sharma, Abhay Omprakash Shirale, Ashis Kumar Biswas, Ashok Kumar Patra.

**Validation:** Satish Bhagwatrao Aher, Madan Lal Reager.

**Visualization:** Priya Gurav Pandurang.

**Writing – original draft:** Chetan Kumar Dotaniya, Mohan Lal Dotaniya, Madan Lal Reager, Ramesh Chandra Sanwal, Rajesh Kumar Doutaniya, Manju Lata.

**Writing – review & editing:** Mohan Lal Dotaniya, Madan Lal Reager, Ramesh Chandra Sanwal, Manju Lata.

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
