## [Decision Letter · Decision Letter 0]

25 Aug 2021

PONE-D-21-15738

Chickpea (Cicer arietinum L.) Performance in a Maize-Chickpea Sequence with Various Integrated Nutrient Modules in a Central Indian Vertisol

PLOS ONE

Dear Dr. Dotaniya,

Thank you for submitting your manuscript to PLOS ONE. After careful consideration, we feel that it has merit but does not fully meet PLOS ONE’s publication criteria as it currently stands. Therefore, we invite you to submit a revised version of the manuscript that addresses the points raised during the review process.

We look forward to receiving your revised manuscript.

Kind regards,

Paulo H. Pagliari

Academic Editor

PLOS ONE

Journal Requirements:

Reviewers' comments:

Reviewer's Responses to Questions

**Comments to the Author**

1. Is the manuscript technically sound, and do the data support the conclusions?

Reviewer #1: Yes

2. Has the statistical analysis been performed appropriately and rigorously? 

Reviewer #1: Yes

3. Have the authors made all data underlying the findings in their manuscript fully available?

Reviewer #1: Yes

4. Is the manuscript presented in an intelligible fashion and written in standard English?

Reviewer #1: Yes

5. Review Comments to the Author

Reviewer #1: this study is a interesting research and it has been done in a good manner. The manuscript has been written in a good manner.

one comment is that the legends of the figures included complete information.

6. PLOS authors have the option to publish the peer review history of their article (what does this mean?). If published, this will include your full peer review and any attached files.

Reviewer #1: No

---

## [Author Response · Author response to Decision Letter 0]

24 Dec 2021

Revised MS as per the direction of EIC and Reviewer comments. A letter is also attached with this submission.

---

## [Editor Report · Decision Letter 1]

4 Jan 2022

Performance of Chickpea (Cicer arietinum L.) in Maize-Chickpea Sequence under Various Integrated Nutrient Modules in a Vertisol of Central India

PONE-D-21-15738R1

Dear Dr. Dotaniya,

We’re pleased to inform you that your manuscript has been judged scientifically suitable for publication and will be formally accepted for publication once it meets all outstanding technical requirements.

Kind regards,

Paulo H. Pagliari

Academic Editor

PLOS ONE
---

## [Editor Report · Acceptance letter]

2 Feb 2022

PONE-D-21-15738R1 

Performance of Chickpea (*Cicer arietinum* L.) in Maize-Chickpea Sequence under Various Integrated Nutrient Modules in a Vertisol of Central India 

Dear Dr. Dotaniya:

I'm pleased to inform you that your manuscript has been deemed suitable for publication in PLOS ONE. Congratulations! Your manuscript is now with our production department. 

Kind regards, 

on behalf of

Dr. Paulo H. Pagliari 

Academic Editor

PLOS ONE